# Changes in Workplace Choice Reasons and Individual Influencing Factors for Nurses Across Job Changes: Cross-Sectional Observational Study

**DOI:** 10.3390/nursrep15020058

**Published:** 2025-02-06

**Authors:** Yukari Hara, Kasumi Sato, Hideyuki Hirayama, Satomi Ito, Junko Omori

**Affiliations:** 1Graduate School of Medicine, Tohoku University, 2-1, Seiryo-machi, Aoba-ku, Sendai 980-8575, Japan; 2Graduate School of Nursing, St. Luke’s International University, 10-1 Akashicho, Chuo-ku, Tokyo 104-0044, Japan

**Keywords:** nurses, workplace, job change, career choices, career mobility, reason for choosing workplace, work values, work motivation

## Abstract

**Background/Objectives:** The global nursing shortage encompasses issues such as the uneven nurse distribution in low- and middle-income countries, nurse outflow to other countries, and nurse retirement in developed countries, necessitating effective retention strategies. Our objective was to clarify the changes in facility selection reasons among Japanese nurses after changing jobs and the personal attributes influencing facility selection. **Methods**: In January 2023, an online survey was conducted among licensed practical nurses, registered nurses, midwives, and public health nurses in Japan. The survey items included personal attributes (age and sex), information on ≤five employment facilities, and facility selection reasons. The variations in facility selection reasons by facility number were analyzed using a generalized linear model with a binomial distribution. A logistic regression analysis was conducted with personal attributes and reasons for workplace selection as the independent and dependent variables, respectively. **Results**: In total, 721 participants were included; 90.8% selected hospitals as their first place of employment. As nurses changed jobs, they increasingly selected non-hospital facilities, including nursing homes, nursery schools, and government agencies. With multiple job changes, the facility selection reasons included “good location for commuting”, “good salary”, and “convenient working style”. Among personal attributes, “age”, “sex”, “age at employment at the facility”, “educational background”, “number of children”, and “living alone” influenced workplace choice reasons. **Conclusions**: Considering the study results, country-specific demographic trends, medical policy changes, and nursing-shortage-related causes, medical facility managers and policymakers should devise appropriate employment conditions and develop recruitment strategies, especially for situations with severe nursing shortages. Nurses can learn from the career choices of others to manage their own careers.

## 1. Introduction

The global nursing shortage has worsened, and the COVID-19 pandemic has increased nurses’ intentions to leave, making nurse retention more difficult [1]. Nursing shortages are more severe in low- and middle-income countries. Southeast Asia is facing a shortage of medical personnel to meet the growing demand for medical care due to rapid aging as well as the concentration of medical personnel in urban areas and labor shortages in rural areas [2]. Similarly, sub-Saharan Africa is facing a serious shortage of medical personnel owing to a rapid growth in population, the concentration of disease burden due to an increase in chronic diseases and infectious diseases such as malaria and tuberculosis, a shortage of medical resources, and the outflow of medical personnel overseas [3]. Furthermore, in the USA, there is concern about a shortage of nurses after the retirement of older nurses; the UK and Germany are trying to solve the nursing shortage by recruiting staff from Central and Eastern Europe and Asia [4].

To address these nursing shortages, previous efforts have focused on nurses’ intention to leave and turnover from jobs and organizations; however, knowledge has gradually accumulated about job changes [5,6,7] and job preferences related to job selection [8,9]. To increase the number of nurses in an organization, it is necessary to have nurses choose that organization and prevent them from changing their jobs; consequently, it is necessary to clarify the reasons for nurses picking an organization and the factors related to nurses changing jobs. Clarifying the types of nurses who select particular organizations and the underlying reasons for this selection could lead to nurses staying with the organization, eliminating regional and professional imbalances and ultimately resolving the global shortage of nurses.

The phenomenon of changing jobs refers to the process of leaving one organization and finding employment at another organization. Regarding the first half of the job-changing process, i.e., leaving an organization, the factors that affect nurses’ intention to leave and turnover rates have been elucidated. A meta-review found that turnover among nurses in adult nursing is associated with personal, educational, work-related, regional, and national factors, and personal factors in particular included demographic characteristics such as sex, age, marital status, and family economic situation [10]. Another review identified nine areas affecting turnover among hospital nurses: nursing leadership and management, education and career advancement, organizational (work) environment, staffing levels, professional issues, support at work, personal influences, demographic influences, and financial remuneration [11]. However, most of these past studies are cross-sectional observational studies of nurses who remain in their organizations and thus lack data on whether those with a high intention to leave do so and on their reemployment after leaving (i.e., changing jobs).

Several studies have focused on job changes among nurses. One study examined midwives and obstetric nurses who graduated from educational institutions in Mali in 2005, 2010, and 2015, revealing career shifts from the private to the public sector, rural to urban areas, and volunteer/unpaid positions to public service through recruitment competitions [7]. A Norwegian study on the clinical fields of two graduate cohorts (2001 and 2003) during their first 10 years of employment [5] found that elderly care in nursing homes or at home was the most common field pursued, reflecting the career expectations of nursing students. A Korean study clarified employment locations at graduation, four months after employment, and four years after graduation at five nursing schools, discovering that nurses’ job changes were related to their grades as students, the type of hospital, their satisfaction with their first job, and their current salary [6].

Studies focusing on nurses’ job changes are rare, and the reasons for choosing facilities during each job change remain unclear. To elucidate nurses’ job changes and related factors, it is necessary to obtain the job change histories and other variables of many participants both cross-sectionally and longitudinally, which requires substantial funds and effort, making research challenging. Additionally, these studies suggest that nurses’ job changes are influenced by the demographic, medical, economic, and employment situations of nurses in each country; therefore, research in various regions is necessary.

In Japan, few studies have examined job changes and related factors among nurses. Many people want to resign owing to life events such as marriage and childcare, with many potential nurses between the ages of 35 and 39 [12]. Furthermore, 11.7% of nurses have changed jobs five or more times [13]. However, to the best of our knowledge, no studies have elucidated the personal factors that influence Japanese nurses’ job change histories or the reasons for choosing a facility with each job change. Therefore, this study was designed to elucidate the personal attributes that influence nurses’ job change histories in Japan and the reasons for choosing a facility each time they change jobs. By clarifying nurses’ job change histories, reasons for selecting facilities, and related personal factors, it may be possible to derive insights that would contribute to addressing the global nursing-shortage-related challenge of uneven nurse distribution across regions and specialties. This exploratory cross-sectional observational study is expected to guide future large-scale cohort studies, and ultimately, this understanding may improve nurse recruitment and retention.

## 2. Materials and Methods

### 2.1. Research Design

This cross-sectional observational study aimed to clarify the career change history of nurses in Japan, targeting those with nursing qualifications (registered nurses, midwives, public health nurses, and licensed practical nurses) who have a history of job changes and are currently working in the field of nursing. Additionally, this quantitative observational study used the Strengthening the Reporting of Observational Studies in Epidemiology (STROBE^®^) guidelines for reporting.

### 2.2. Setting and Sample

In January 2023, a web survey was conducted among 2000 qualified assistant nurses, registered nurses, midwives, and public health nurses in Japan. The survey was outsourced to a Japanese company that conducted academic research. The researcher created a list of questions edited by a contractor for the web survey. First, individuals with one of four qualifications (registered nurses, midwives, public health nurses, and licensed practical nurses) were screened from a nationwide panel of approximately 5 million Japanese people managed by the contractor. A web survey was conducted among those who met the screening criteria. The survey was closed once 2000 participants responded. The survey period was two days: 23–24 January 2023.

Upon notification of a web survey request online by the contractor, the participants initially answered the screening items. If they met the screening criteria, an explanatory document for this study was displayed, and if they agreed to participate, they checked a checkbox to indicate their consent. After providing consent, they answered the displayed survey items. All participants responded online, and the survey consisted of a single web survey that took approximately 15 min to complete.

This study was part of the initial survey of a project aimed at clarifying the type of occupational mobility of people with four qualifications and the relationship between their work values and psychosocial variables. The project planned to use a machine learning program, and the sample size was set at 2000 based on consultation with the data scientist who developed the program. To meet project requirements, the overall project included individuals with the four qualifications, even those not currently working or on maternity/childcare leave.

The eligibility criteria for participants in this study were as follows: (1) having changed jobs at least once; (2) having changed jobs across up to five facilities since their first employment facility; and (3) currently working as a licensed practical nurse, registered nurse, midwife, or public health nurse. Those who did not respond in Japanese were excluded from this study. Of the 2000 respondents, 721 were included in the analysis. The exclusion process was as follows: From the 2000 respondents, 510 participants who were not currently working or who did not enter their current workplace information were excluded. Next, 270 participants who were not currently working as licensed practical nurses, registered nurses, midwives, or public health nurses were further excluded. Furthermore, 499 participants who had not changed jobs were excluded, leaving 721 participants for the final analysis.

### 2.3. Measures

#### 2.3.1. Demographic Variables

We collected personal attributes, including age at the time of the survey, sex, marital status, whether they had children, whether they had family caregivers, whether they lived alone, and their educational background.

#### 2.3.2. Variables Related to Job Change

A previous survey in Japan found that 27.5% of nurses had worked in one facility, 10.6% had worked in up to four facilities, and only 7.1% had worked in five facilities [13]. Therefore, this study collected detailed information from the first to the fifth facilities. Facility types included hospitals, clinics, and others (e.g., nursing care facilities, visiting nurses, nursery schools/kindergartens, health centers, general companies, and nursing training institutions, such as universities or vocational schools). If a hospital was selected, the number of beds and type (highly acute, acute, recovery, or chronic) were also recorded.

Ten workplace selection reasons were set based on a previous Japan survey [13] and recent studies on nurses’ job preferences [8,9], and participants selected the option that best represented their reason for choosing a workplace from the following: “Good education and career advancement system”, “good employee benefits”, “convenient location for commuting”, “well-known”, “recommended by family, acquaintances, or teachers”, “good salary”, “convenient working style”, “I can demonstrate my abilities”, “I can do what I want”, and “atmosphere such as human relations in the workplace seems good”. Multiple selections were allowed for these items.

### 2.4. Statistical Analyses

First, descriptive statistics were calculated for each item, including personal attributes, workplace classification, and reasons for choosing a workplace. Cross-tabulations were performed to analyze the reasons for choosing a workplace based on the number of job changes and personal attributes.

Next, as each participant provided reasons for choosing up to the fifth facility, a generalized linear mixed model (GLMM) with a binomial distribution was used for analysis. The 10 reasons for choosing a workplace were set as dependent variables. The number of employment facilities was used as a fixed effect, and personal ID was set as a random effect.

Finally, univariate and multivariate logistic regression analyses were performed with the reason for choosing the workplace as the dependent variable. To examine the influence of multicollinearity in the logistic regression analysis, the distribution of personal attributes was confirmed. Pearson’s product-moment correlation coefficient was calculated between age at the time of the survey and age at the time of employment at the facility.

Because there is little previous research on nurse job changes, the independent variables used in the logistic regression analysis were set based on two systematic reviews related to nurse turnover [10,11]. Specifically, the independent variables were the following: age at the time of the survey, age at the time of employment at each facility, sex, marital status, number of children, family caregivers, living alone, and educational background. For marital status and number of children, if their distribution was similar based on descriptive statistics, only the number of children was included in the analysis. According to a survey of nurses in Japan, parenting was a more common reason than marriage for quitting jobs [12]. In the logistic regression analysis, a univariate analysis was performed first. A multiple logistic regression analysis was subsequently conducted, incorporating all variables.

In the multiple logistic regression analysis, it was assumed that the age at the time of employment at the first facility, number of children, family caregiver status, and living alone status may differ from the time of the survey. Therefore, when performing multiple logistic regression analysis for choosing the first facility as the dependent variable, only the age at the time of the survey, age at employment at the facility, sex, and educational background were included as independent variables.

In Japan, many individuals leave their jobs because of life events such as marriage and childcare [12]; therefore, it was assumed that personal attributes at employment for the second or subsequent facility (such as marriage/childbirth) would align with the time of the survey. Therefore, in the multiple logistic regression analysis with reasons for choosing the second or subsequent facility as the dependent variable, the independent variables included age at the time of the survey, age at the time of employment at each facility, sex, number of children, caregiver status, living alone, and educational background. The correlation coefficient between age at the time of facility selection and age at the time of the survey was confirmed. When the absolute value of the correlation coefficient exceeded 0.8, indicating potential multicollinearity, the age at the time of facility employment was excluded from the analysis [14].

In addition, previous studies indicate that a sample size 10 times that of the number of independent variables is needed for multiple logistic regression analysis [15]. In this study, multiple logistic regression analysis was conducted using the reason for choosing a facility at the time of employment as the dependent variable. Only participants with a sample size 10 times that of the number of explanatory variables were included in the analysis. JMP Pro (version 17) was used for statistical analysis, with a significance level set at 5% on both sides.

### 2.5. Institutional Review Board Statement

This study was conducted in accordance with the ethical guidelines for medical and health research involving human participants in Japan and the 1995 Helsinki Declaration (revised in Edinburgh 2000) and was approved by the Tohoku University Graduate School of Medicine Ethics Committee (number: 2022-1-788; approval date: 20 December 2022). Participants received an online document outlining this study’s content, research objectives, and research methods, which emphasized their voluntary participation in this study, assured them of no penalties for non-participation, guaranteed anonymity, and informed them that the analyzed data would be presented at academic conferences and published in research papers.

## 3. Results

### 3.1. Descriptive Statistics of Participants’ Personal Attributes

The participants’ attributes are listed in Table 1: the descriptive statistics for the 721 participants included in the analysis are provided in the column “1st facility/2nd facility”. The mean age of the participants at the time of the survey was 43.3 ± 9.47 years, and 633 (87.8%) participants were female. An example of a personal attribute whose proportions increased with the number of job changes is marital status (married). Additionally, examples of personal attributes that saw a decrease in percentages are living alone (yes) and educational background (bachelor’s or graduate school in nursing).

In terms of workplace classification, 90.8% of participants first worked at a hospital. With each job change, the participants increasingly selected facilities other than hospitals, such as nursing homes, nurseries, and government agencies. Additionally, participants preferred small- to medium-sized hospitals over large ones with each job change. Regarding hospital classification, participants selected highly acute care hospitals less frequently and chronic care hospitals more frequently with each job change.

### 3.2. Reasons for Choosing a Workplace for Each Job Change

Table 2 presents descriptive statistics for reasons for choosing a workplace by the number of employment facilities and the results of an analysis using a GLMM. Table 3 displays the statistical values from the GLMM, showing the influence of the number of employment facilities on each reason for choosing a workplace. The proportion of participants selecting “convenient location for commuting”, “good salary”, and “convenient working style” was significantly higher with an increased number of employment facilities (*p* < 0.001, *p* < 0.01, *p* < 0.001, respectively). Conversely, significantly fewer participants selected “good education and career advancement system”, “good employee benefits”, and “well-known” as their reasons for choosing a workplace (all *p* < 0.001).

### 3.3. Relationship Between Reasons for Workplace Selection and Personal Attributes by Number of Employment Facilities

Table 4 and Table 5 present the results of univariate and multivariate logistic regression analyses. The Pearson product-moment correlation coefficients between the age at the time of the survey and the age at employment at each facility were 0.09 (*p* = 0.013) for the first facility, 0.38 (*p* < 0.001) for the second, 0.50 (*p* < 0.001 for the third, 0.63 (*p* < 0.001) for the fourth, and 0.79 (*p* < 0.001) for the fifth, with no absolute values exceeding 0.8. In addition, as shown in Table 1, the percentage of married people and that of people with children was almost the same, leading to the exclusion of marital status as an independent variable. Therefore, for the first facility, the independent variables included age at the time of the survey, age at the time of employment, sex, and educational background. From the second facility onwards, the number of children, family caregivers, and whether they lived alone were added as additional independent variables. As there were seven independent variables, only reasons for facility selection with *n* = 70 or more were analyzed, noting that the fifth facility did not meet the *n* = 70 criteria for any item.

The results of the multiple logistic regression analysis indicated that for the first facility “good education and career advancement system” was associated with age at the time of the survey and educational background, chosen by younger nurses (odds ratio [OR] = 0.97) and those with a university or graduate school education (OR = 2.41). “Good employee benefits” were also associated with age at the time of the survey and were chosen by younger nurses (OR = 0.98). “Well-known” was associated with educational background, chosen by nurses with a university or graduate school education (OR = 2.20). “Recommended by family, acquaintances, or teachers” was associated with age at the time of the survey, chosen by older nurses (OR = 2.95). “Convenient working style” was associated with age at the time of employment at the facility, chosen by older nurses (OR = 1.04). Factors associated with “I can demonstrate my abilities” included age at the time of the survey and educational background, chosen by older nurses (OR = 1.07) and those with a university or graduate school degree (OR = 2.18). “I can do what I want to do” was associated with educational background, chosen by nurses who had a university or graduate school degree (OR = 2.04).

For the second facility, age at the time of the survey was associated with “good employee benefits”, chosen by younger nurses (OR = 0.97). Living alone was associated with “convenient location for commuting”, with nurses who lived alone not selecting this reason (OR = 0.61). Age at the time of the survey was associated with “recommended by family, acquaintances, or teachers”, chosen by older nurses (OR = 1.03). Age at the time of the survey was associated with “good salary”, chosen by younger nurses (OR = 0.97). Age at the time of the survey, age at the time of employment at the facility, and number of children were associated with “convenient working style”, chosen by younger nurses (OR = 0.96), older nurses at the time of employment (OR = 1.06), and those with children (OR = 1.72). Factors associated with “I can demonstrate my abilities” included age at the time of the survey and educational background, chosen by older nurses (OR = 1.04) with a university or graduate school degree (OR = 3.41). Living alone was the factor associated with “I can do what I want”, chosen by nurses who lived alone (OR = 1.90).

For the third facility, sex was associated with “convenient location for commuting”, chosen by women (OR = 2.65). Sex and living alone were associated with “convenient working style”, chosen by women (OR = 9.00) and not by nurses living alone (OR = 0.54). Educational background was associated with “I can do what I want”, chosen by nurses with university or graduate degrees (OR = 2.35).

For the fourth facility, sex was associated with a “convenient location for commuting”, chosen by women (OR = 3.03).

To summarize the results in Table 4 and Table 5, the personal attributes related to reasons for choosing a facility were age at the time of the survey, educational background, age at the time of employment at the facility, living alone, number of children, and sex. Educational background and age at the time of the survey influenced the reasons for choosing a facility, regardless of the number of times they had changed jobs, but sex had an influence when changing jobs to their third or fourth facility.

## 4. Discussion

There have been very few studies focusing on nurses’ job changes. It is unclear why nurses choose a facility, why they change jobs, and how their personal factors influence these reasons. This study examined the relationship between workplace selection and personal factors based on the number of employment facilities, focusing on nurses who have changed jobs. The results showed that the reasons for choosing a facility, such as “good location for commuting”, “good salary”, and “convenient working style”, increased with repeated job changes. Furthermore, this study clarified that personal attributes, including “age at the time of the survey”, “sex”, “age at the time of employment at the facility”, “educational background”, “number of children”, and “living alone”, influenced the reasons for choosing a workplace. By clarifying the reasons underlying nurses’ workplace selection and the related personal factors reported in this study, medical institutions and policymakers will be able to consider more appropriate employment conditions and recruitment strategies. In addition, by considering each country’s demographic trends, trends in medical policies, and the causes of nurse shortages, the findings of this study may contribute to addressing the global nursing shortage.

### 4.1. Participant Overview

The participants’ average age at employment was 29.5 ± 6.84 years at the second facility, 33.6 ± 7.02 years at the third facility, and 37.7 ± 7.59 years at the fourth facility. The employment age at the second and third facilities is close to the average age of first marriage (29.4 years) [16] and first childbirth (30.7 years) for Japanese women [17], which supports past surveys indicating that life events may influence job changes among nurses in Japan. However, as approximately 40% of the participants were unmarried, these ages may represent turning points in nursing careers.

In this study, 91% of nurses chose hospitals as their first place of employment, which is similar to the 97% reported in a Korean study, indicating that most nurses in Japan and Korea begin their careers in hospitals [6]. In contrast, a Norwegian study found that 63% of nurses first worked in general hospital care and 26% in nursing homes or home elderly care, which differs from the findings in Japan and Korea [5]. Because Japan has one of the highest proportions of elderly people worldwide, it is assumed that there is also a need for newly graduated nurses to work in nursing homes or in-home elderly care; however, nursing education and policies may not be keeping up with the rapid aging of the population [12]. These differences in nursing careers may be caused by shifts in demographics and healthcare policies in each country, and thus, broader healthcare trends need to be considered.

In addition, this study revealed that, as nurses changed jobs, they chose non-hospital facilities such as nursing homes, nurseries, and government agencies, and when changing jobs within hospitals, they opted for small- to medium-sized facilities instead of large ones. Two reasons are speculated to underlie this observation. First, government-led healthcare policies such as the Community Comprehensive Care System have increased the demand for nurses in non-hospital facilities such as visiting nursing stations, and working conditions and compensation are gradually improving [18]. Second, large-sized highly acute care hospitals have a heavy workload and high risk of burnout, while small- to medium-sized chronic care hospitals and non-hospital environments often have flexible working conditions such as part-time work schedules and have been shown to have high job satisfaction among employees [19].

### 4.2. Changes in Workplace Selection Reasons According to Employment History

More than 30% of participants selected a “convenient location for commuting” as a reason for choosing their first workplace, with this percentage increasing at their second and third facilities of employment. Furthermore, although a “convenient working style” was not important at their first workplace (9.4%), it gained significance as they worked at more facilities. These results align with studies of nurses and nurse practitioners in the United States, which highlight preferences for “flexible scheduling” and “a shorter commute time” [9,20], suggesting that flexibility in working conditions and commuting is crucial for nurses seeking long-term employment. In addition, in this study, the number of participants who chose a “good salary” increased significantly at their second and fourth facilities, demonstrating that salary increases become more important as they change jobs. Healthcare facility managers, especially in regions and nursing fields with considerable nursing shortages, can enhance recruitment by offering flexible work arrangements and competitive salaries that are commensurate with career advancement.

However, the importance of a “good education and career advancement system” decreased with each job change. This result was similar to that of a Norwegian study [5] that indicated that many nurses prioritize career prospects over status when choosing their careers. The participants in this study increasingly selected non-hospital facilities, such as nursing homes, nursery schools, and government agencies, with each job change. They tended to prefer small- and medium-sized hospitals, suggesting a shift toward medical institutions more aligned with their specialties rather than large organizations’ education and career advancement systems. The proportion of participants who chose “good employee benefits” and “well-known” significantly decreased from the first to the second facility, indicating that experience in medical institutions made commuting time and working style more important than welfare benefits. This suggested that nurses are choosing medical institutions that benefit their needs, rather than relying on reputation.

### 4.3. Relationship Between Reasons for Choosing Workplaces Based on the Number of Employment Facilities and Personal Attributes

This study revealed that “age at the time of the survey”, “sex”, “age at the time of employment at the facility”, “educational background”, “number of children”, and “living alone” influence workplace selection based on the number of employment facilities.

Regarding “age at the time of the survey”, younger nurses prioritized a “good education and career advancement system” at their first facility, “good employee benefits” at both the first and second facilities, and a “good salary” and “convenient working style” at the second facility, revealing extrinsic motivation for their choices. In contrast, older nurses prioritized “recommendation by family, acquaintances, or teachers” and “I can demonstrate my abilities” at both the first and second facilities, suggesting generational differences in the work values of nurses. These results align with previous research indicating that Generation X (born 1965–1980) nurses prioritize intrinsic values more than Generation Y (born 1981–1996) and Generation Z (born after 1996) nurses [21], who emphasize work–life balance as the generational cohort becomes younger [22]. Addressing the different values across generations and compensating for unmet needs may help reduce turnover in organizations.

In addition, nurses who are older at the time of employment are more likely to choose a “convenient working style” at both the first and second facilities. In Japan, approximately 20% of entrants to three-year nursing schools have previous working experience outside nursing, including roles related to raising children [23]. In this study, nurses with children preferred a “convenient working style” at the second facility, suggesting that those with family responsibilities value flexibility in working style. Conversely, nurses living alone significantly valued the ability to “do what I want” at the second facility, with an OR of 0.61 for choosing a “convenient location for commuting” at the second facility and an OR of 0.54 for choosing a “convenient working style” at the third facility, indicating lower importance placed on these factors. Nurses who live alone may prioritize their preferred nursing roles over location or working style. Regarding “sex”, the third and fourth facilities were associated with the preference for a “convenient location for commuting”, whereas the third facility was associated with a “convenient working style”. These results revealed that women and those with family roles prioritize flexibility in commuting and working hours. Although sex disparities in childcare and housework are gradually being eliminated in the United States [24], considerable sex disparities remain in many countries, including Japan and Italy, and the COVID-19 pandemic has further increased the burden of housework and childcare on women [25,26]. Japan’s 2024 Gender Gap Index ranks 118th out of 146 countries in the world, dropping to 120th in economic activity indicators, including the male–female labor force participation rate [27], making it one of the countries with the largest sex disparities in the world. These contexts may influence nurses’ choices regarding their workplaces.

Finally, regarding “educational background”, nurses with a university or graduate-level education in nursing chose a “good education and career advancement system” and “well-known” at their first facility and “I can demonstrate my abilities” at their first and second facilities. These nurses also valued “I can do what I want” at the first and third facilities. This suggested that nurses with higher education levels prioritized workplaces that emphasized intrinsic values, such as career advancement and skill improvement. According to the self-determination theory, intrinsic motivation leads to positive outcomes, such as improved performance, when basic psychological needs for autonomy, competence, and relatedness are supported [28]. In countries such as Japan [12], the USA, and China [29] where basic nursing education is provided outside of undergraduate and graduate schools, it is necessary to support the basic psychological needs of nursing students in junior colleges, nursing schools, and vocational schools. Such support can foster intrinsic motivation, encouraging career advancement and skill development once they enter the workforce.

### 4.4. Implications for Individual Nurse and Healthcare Organization Practice, Policy Implications, and Future Research

This study surveyed Japanese nurses regarding job changes; the results of this study may help address the global nursing shortage by examining the impact on individual nurses, medical institution practices, and policies.

First, for health facility managers and policymakers, offering flexible work arrangements and competitive salaries commensurate with career advancement may enhance nurse recruitment and retention, especially in regions and fields with a severe nursing shortage. The current study results showed that flexible work arrangements and salaries were emphasized as nurses progressed in their careers. As a specific example, Southeast Asia has a high concentration of medical workers in urban areas, where salaries are higher, while rural areas are understaffed [2]. The USA can be used as a reference, where employers offer nurses high bonuses, free housing, and subsidies for children’s school fees [30]. However, from a management perspective, it is not realistic for medical institutions to implement such measures on their own, and thus, this needs to be considered as part of policy measures at the country level.

Even among developed countries, countries with large gender disparities such as Japan and Italy need support to help female workers continue with their employment. The current study results showed that sex, the number of children, and living alone influenced the reasons for choosing a workplace. Medical facility managers and policymakers need to improve support for nurses such as providing on-site childcare facilities and housekeeping services.

In addition, support for foreign nurses to settle in is necessary. In countries with declining birth rates and aging populations, it is necessary to accept foreign nurses because domestic nurses alone cannot care for the volume of older adults. Although countries with closed cultures such as Japan have considerable challenges in accepting nursing staff from overseas, support for nurse education is provided, and the pass rate for the national nursing examination for foreign nurses is reported to be increasing [31]. Regarding Germany, the Shandong International Nursing Training Center in China provides nurses with a training program that includes language and intercultural training to prepare them to work in Germany [4]. Because the current study focused on nurses in Japan, job transfers to other countries were not evaluated, but globally, nurse mobility is common, and support for retention is important [32].

It is also necessary to prevent the outflow of nurses to other countries. As observed in this study, nurses move to places with more attractive working conditions and salaries. Globally, nurses are migrating from low-income and middle-income countries to high-income countries; consequently, the International Council of Nurses has emphasized that it is necessary to stop the outflow of nurses from countries with serious medical-worker shortages and weakened medical infrastructure because this will worsen deep-rooted health disparities worldwide [33]. High-income countries must be careful about accepting nurses from countries with serious nurse shortages, and it is recommended that low- and middle-income countries provide support for nurse education and labor costs. In sub-Saharan Africa, the education of nurses is not keeping up with the growth in population. In such regions, in addition to policies, support from other countries is needed to review the educational level of domestic nurses, increase the number of educational facilities, improve education through subsidies, and secure the status and income of nurses.

Finally, as noted in this study, individual nurses can use the knowledge of others’ choices to help manage their own careers because it is necessary to understand the current situation of the global nursing shortage and consider where and how to work as a nurse.

As a suggestion for future research, because the participants in this study were limited to nurses working in Japan, similar studies should be conducted in various countries and regions in the future to achieve wider applicability. Global longitudinal research is essential because it is common for nurses to move abroad. Because individual researchers or small research teams may find it difficult to perform such large-scale studies, it is recommended to conduct international surveys as a nursing policy. In addition, although this study only dealt with personal attributes as influencing factors for workplace selection reasons, other factors that have been identified to cause nurses to leave their jobs, such as regional characteristics and facility characteristics (e.g., hospital/non-hospital and hospital size), should be considered.

### 4.5. Limitations

This study maintained quality standards by using the STROBE statement checklist. However, several limitations remained.

First, the primary limitation was that for some participants the survey and job change timelines were far apart. Therefore, personal attributes such as marital status, the number of children, and family caregiver status may have differed between the time of the survey and the time of the job change, preventing the precise analysis of the relationship between personal attributes and job changes. In addition, as a cross-sectional observational study, participants recalled their job changes, leading to potential recall bias in the results. To address these limitations and elucidate the relationship between workplace choices and personal attributes, a cohort study that follows nurses from the time they are hired is necessary.

Second, the web survey could have introduced bias. The web survey could be completed on different devices, such as smartphones, tablets, and personal computers; thus, differences in screen size and interface may have influenced survey participation and response behavior, especially for older nurses. In addition, because this web survey was voluntary, participation may have been biased toward those who were interested in the research topic, creating a risk that the data of people with an interest or opinion on the study topic were over-represented.

## 5. Conclusions

This study analyzed the relationship between workplace choice reasons and personal factors among Japanese nurses who have changed jobs, based on the number of employment facilities. The results showed that the reasons for choosing a facility, such as a “good location for commuting”, “good salary”, and “convenient working style”, increased with repeated job changes. In addition, it was clarified that personal attributes such as “age at the time of the survey”, “sex”, “age at the time of employment at the facility”, “educational background”, “number of children”, and “living alone”, influenced workplace choices. Considering the study results, along with each country’s demographic trends, changes in medical policies, and causes of the nurse shortage, medical facility managers and policymakers can devise appropriate employment conditions and develop recruitment strategies, especially in areas and nursing fields with severe nurse shortages in low- and middle-income countries. Even in developed countries with large gender disparities, support is needed to help female workers continue with their employment. Nurses can use the knowledge of the career choices of others to help manage their own careers. In this way, examining the practical and policy impact on individual nurses and medical institutions based on the results of this study may help develop solutions to address the global nursing shortage.

## Figures and Tables

**Table 1 nursrep-15-00058-t001:** Personal attributes/workplace classification.

	1st Facility	2nd Facility	3rd Facility	4th Facility	5th Facility
	*n* = 721		*n* = 420	*n* = 210	*n* = 82
Number of people still working at the facility at the time of the survey (*n*, % of total)	0	301 (41.7)	210 (29.1)	128 (17.8)	82 (11.4)
	*n* (%)		*n* (%)	*n* (%)	*n* (%)
Age at time of study (mean ± SD ^†^)	43.3 ± 9.47		45.0 ± 8.82	46.4 ± 8.45	47.5 ± 8.73
20–29 years	65 (9.2)		16 (3.8)	13 (6.2)	2 (2.4)
30–39 years	182 (25.2)		92 (22.0)	53 (25.2)	13 (15.9)
40–49 years	285 (39.5)		181 (43.1)	87 (41.4)	34 (41.5)
50–59 years	159 (22.0)		113 (26.9)	50 (23.8)	25 (30.5)
60–69 years	30 (4.2)		18 (4.3)	7 (3.3)	8 (9.8)
Age at time of employment at each facility (mean ± SD ^†^)	23.1 ± 5.35	29.5 ± 6.84	33.6 ± 7.02	37.7 ± 7.59	41.6 ± 8.59
Years of work at each facility (mean ± SD ^†^)	5.5 ± 5.42	4.15 ± 4.76	4.6 ± 4.38	3.7 ± 4.52	5.9 ± 5.60
**Sex** Male	88 (12.2)		45 (10.7)	20 (9.5)	8 (9.8)
Female	633 (87.8)		375 (89.3)	190 (90.5)	74 (90.2)
**Marital status** Married	437 (60.6)		256 (61.0)	127 (61.5)	52 (63.4)
Unmarried	260 (36.1)		149 (35.5)	73 (34.8)	25 (30.5)
Other (divorced, widowed, etc.)	24 (3.3)		15 (3.6)	10 (4.8)	5 (6.1)
**Number of children** One or more	435 (60.3)		261 (62.1)	135 (64.3)	52 (63.4)
None	286 (39.7)		159 (37.9)	75 (35.7)	30 (36.6)
**Family caregivers** Yes	28 (3.9)		21 (5.0)	9 (4.3)	2 (2.4)
No	693 (96.1)		399 (95.0)	201 (95.7)	80 (97.6)
**Living alone** Yes	169 (23.4)		95 (22.6)	44 (21.0)	16 (19.5)
No	552 (76.6)		325 (77.4)	166 (79.1)	66 (80.5)
Educational background					
Baccalaureate program (4-year program in nursing) or graduate school in nursing	170 (23.6)		92 (21.9)	43 (20.5)	15 (18.3)
Other (2-year nursing course, 3-year nursing course, 5-year integrated nursing school, licensed practical nurse training school)	551 (76.4)		328 (78.1)	167 (79.5)	67 (81.7)
Hospitals	655 (90.8)	425 (59.0)	205 (48.8)	77 (36.7)	22 (26.8)
Clinics	41 (5.7)	139 (19.3)	87 (20.7)	49 (23.3)	16 (19.5)
Other (nursing care facilities, visiting nurse, nursery school/kindergarten, health center, general company, nursing training institution such as university or vocational school)	25 (3.5)	157 (21.7)	128 (30.5)	84 (40.0)	44 (53.7)
**Hospital size** Large-scale (500 beds or more)	305 (46.6)	86 (20.2)	37 (18.0)	11 (14.3)	3 (13.6)
Medium-scale (100–499 beds)	295 (45.0)	262 (61.6)	127 (62.0)	45 (58.4)	14 (63.6)
Small-scale (less than 100 beds)	55 (8.4)	77 (18.1)	41 (20.0)	21 (27.3)	5 (22.7)
**Hospital classification** Highly acute	166 (25.3)	55 (12.9)	29 (14.1)	7 (9.1)	2 (9.1)
Acute	347 (53.0)	215 (50.6)	97 (47.3)	40 (52.0)	9 (40.9)
Recovery	37 (5.6)	37 (8.7)	15 (7.3)	11 (14.3)	1 (4.5)
Chronic	105 (16.0)	118 (27.8)	64 (31.2)	19 (24.7)	10 (45.5)

^†^ SD: standard deviation.

**Table 2 nursrep-15-00058-t002:** Descriptive statistics and analysis results of reasons for choosing a workplace based on the number of employment facilities.

	Fixed Effects (Number of Facilities)		1st Facility	2nd Facility	3rd Facility	4th Facility	5th Facility
			*n* = 721		*n* = 420	*n* = 210	*n* = 82
Reason for choosing the workplace	*F*-score	*p*-value	*n* (%)		*n* (%)	*n* (%)	*n* (%)
1. Good education and career advancement system	55.2	***	292 (40.5)	**104 (14.4)**	**38 (9.0)**	14 (6.7)	1 (1.2)
2. Good employee benefits	13.8	***	209 (29.0)	**129 (17.9)**	55 (13.1)	30 (14.3)	8 (9.8)
3. Convenient location for commuting	7.9	***	227 (31.5)	**275 (38.1)**	**186 (44.3)**	98 (46.7)	37 (45.1)
4. Well-known	15.1	***	117 (16.2)	**40 (5.5)**	23 (5.5)	12 (5.7)	5 (6.1)
5. Recommended by family, acquaintances, or teachers	2.1	0.073	130 (18.0)	118 (16.4)	65 (15.5)	22 (10.5)	9 (11.0)
6. Good salary	3.8	**	93 (12.9)	**129 (17.9)**	65 (15.5)	**48 (22.9)**	18 (22.0)
7. Convenient working style	41.0	***	68 (9.4)	**222 (30.8)**	**149 (35.5)**	**93 (44.3)**	38 (46.3)
8. I can demonstrate my abilities	4.5	**	53 (7.4)	70 (9.7)	48 (11.4)	36 (17.1)	8 (9.8)
9. I can do what I want	0.4	0.780	124 (17.2)	135 (18.7)	72 (17.1)	44 (21.0)	17 (20.7)
10. Atmosphere such as human relations in the workplace seems good	0.1	0.982	51 (7.1)	97 (12.1)	41 (9.8)	29 (13.8)	13 (15.9)

** *p* < 0.01, *** *p* < 0.001. Note 1: bold numbers indicate a significant change in the proportion of reasons for choosing a workplace based on the previous number of employment facilities. Note 2: statistical indicators for the proportion of reasons for choosing a workplace are shown in Table 3. Note 3: In the analysis of the generalized linear mixed model, only the intercept of the fixed effect for “8. I feel I can demonstrate my abilities” was significant. There was no significant increase or decrease in the proportion of reasons for choosing a workplace based on the number of facilities.

**Table 3 nursrep-15-00058-t003:** Generalized linear mixed model results.

1. Good Education and Career Advancement System	Estimate	Standard Error	*t*-Value	*p*-Value (Prob > |t|)		95% Lower	95% Upper
Intercept	−0.389	0.080	−4.87	<0.0001	***	−0.546	−0.233
Number of facilities [2-1]	−1.412	0.131	−10.78	<0.0001	***	−1.669	−1.155
Number of facilities [3-2]	−0.528	0.202	−2.61	0.009	**	−0.924	−0.132
Number of facilities [4-3]	−0.316	0.327	−0.97	0.334		−0.957	0.326
Number of facilities [5-4]	−1.670	1.045	−1.6	0.110		−3.720	0.380
2. Good employee benefits							
Intercept	−0.928	0.090	−10.36	<0.0001	***	−1.103	−0.752
Number of facilities [2-1]	−0.650	0.129	−5.02	<0.0001	***	−0.904	−0.396
Number of facilities [3-2]	−0.349	0.179	−1.95	0.051		−0.699	0.002
Number of facilities [4-3]	0.095	0.251	0.38	0.704		−0.397	0.588
Number of facilities [5-4]	−0.384	0.434	−0.89	0.376		−1.235	0.466
3. Convenient location for commuting							
Intercept	−0.807	0.087	−9.31	<0.0001	***	−0.978	−0.637
Number of facilities [2-1]	0.305	0.113	2.7	0.007	**	0.084	0.527
Number of facilities [3-2]	0.301	0.129	2.33	0.020	*	0.048	0.555
Number of facilities [4-3]	0.109	0.177	0.62	0.538		−0.238	0.456
Number of facilities [5-4]	−0.023	0.275	−0.08	0.935		−0.563	0.518
4. Well-known							
Intercept	−1.676	0.107	−15.68	<0.0001	***	−1.885	−1.466
Number of facilities [2-1]	−1.210	0.193	−6.28	<.0001	***	−1.588	−0.833
Number of facilities [3-2]	−0.020	0.272	−0.07	0.941		−0.553	0.513
Number of facilities [4-3]	0.041	0.371	0.11	0.911		−0.686	0.768
Number of facilities [5-4]	0.123	0.556	0.22	0.825		−0.967	1.214
6. Good salary							
Intercept	−1.935	0.114	−16.91	<0.0001	***	−2.160	−1.711
Number of facilities [2-1]	0.390	0.148	2.63	0.009	**	0.099	0.681
Number of facilities [3-2]	−0.173	0.169	−1.03	0.305		−0.503	0.158
Number of facilities [4-3]	0.475	0.217	2.19	0.029	*	0.050	0.900
Number of facilities [5-4]	−0.040	0.321	−0.13	0.900		−0.669	0.589
7. Convenient working style							
Intercept	−2.347	0.133	−17.62	<0.0001	***	−2.609	−2.086
Number of facilities [2-1]	1.501	0.153	9.81	<0.0001	***	1.201	1.802
Number of facilities [3-2]	0.267	0.136	1.96	0.0497	*	0.000	0.534
Number of facilities [4-3]	0.391	0.182	2.15	0.032	*	0.034	0.748
Number of facilities [5-4]	0.060	0.279	0.21	0.831		−0.488	0.607
8. I can demonstrate my abilities							
Intercept	−2.727	0.156	−17.48	<0.0001	***	−3.033	−2.421
Number of facilities [2-1]	0.326	0.197	1.65	0.098		−0.060	0.712
Number of facilities [3-2]	0.211	0.210	1	0.317		−0.202	0.623
Number of facilities [4-3]	0.504	0.257	1.96	0.050		−0.001	1.009
Number of facilities [5-4]	−0.592	0.446	−1.33	0.184		−1.467	0.282

* *p* < 0.05, ** *p* < 0.01, *** *p* < 0.001.

**Table 4 nursrep-15-00058-t004:** Relationship between reasons for choosing workplace and personal attributes (1st and 2nd facilities).

Number of Employment Facilities	1st Facility (*n* = 721)								2nd Facility (*n* = 721)							
Number of job changes	0								1							
	Univariate				Multivariate				Univariate				Multivariate			
	Odds ratio	Lower 95%	Upper 95%	*p*-value	Odds ratio	Lower 95%	Upper 95%	*p*-value	Odds ratio	Lower 95%	Upper 95%	*p*-value	Odds ratio	Lower 95%	Upper 95%	*p*-value
1. Good education and career advancement system	*n* = 292								*n* = 104							
Age at time of survey	**0.96**	**0.95**	**0.98**	**<0.0001**	**0.97**	**0.96**	**0.99**	**0.003**	0.98	0.96	1.00	0.074	0.99	0.96	1.01	0.372
Age at time of employment at facility	1.02	0.99	1.05	0.166	1.03	1.00	1.06	0.050	0.98	0.95	1.01111	0.194	0.98	0.95	1.02	0.346
Sex (female = 1)	0.93	0.59	1.46	0.753	0.97	0.60	1.58	0.907	0.61	0.35	1.08	0.088	0.56	0.31	1.01	0.056
Educational background (university/graduate school = 1)	**2.85**	**2.00**	**4.06**	**<0.0001**	**2.41**	**1.66**	**3.51**	**<0.0001**	1.31	0.82	2.08	0.264	1.15	0.69	1.90	0.589
Number of children (one or more = 1)									0.77	0.50	1.17	0.214	0.70	0.42	1.16	0.164
Family caregivers (yes = 1)									1.30	0.48	3.51	0.599	1.21	0.44	3.30	0.712
Living alone (living alone = 1)									0.75	0.44	1.26	0.275	0.58	0.32	1.04	0.069
2. Good employee benefits	*n* = 209								*n* = 129							
Age at time of survey	**0.97**	**0.95**	**0.99**	**<0.001**	**0.98**	**0.96**	**0.99**	**0.008**	**0.98**	**0.96**	**1.00**	**0.016**	**0.97**	**0.95**	**0.99**	**0.015**
Age at time of employment at facility	0.97	0.94	1.00	0.052	0.98	0.94	1.01	0.173	0.99	0.96	1.02	0.429	1.00	0.97	1.03	0.959
Sex (female = 1)	1.56	0.91	2.66	0.105	1.33	0.76	2.32	0.316	0.90	0.51	1.58	0.710	0.82	0.46	1.47	0.501
Educational background (university/graduate school = 1)	**1.63**	**1.13**	**2.35**	**0.008**	1.33	0.90	1.97	0.156	1.26	0.82	1.94	0.295	1.05	0.66	1.68	0.829
Number of children (one or more = 1)									1.09	0.74	1.61	0.666	1.43	0.87	2.34	0.159
Family caregivers (yes = 1)									0.76	0.26	2.22	0.613	0.77	0.26	2.28	0.632
Living alone (living alone = 1)									0.99	0.63	1.55	0.957	1.10	0.64	1.89	0.719
3. Convenient location for commuting	*n* = 227								*n* = 275							
Age at time of survey	0.99	0.97	1.00	0.145	0.53	0.25	1.12	0.095	**1.02**	**1.00**	**1.03**	**0.027**	1.02	1.00	1.04	0.089
Age at time of employment at facility	0.99	0.96	1.02	0.363	0.68	0.16	2.97	0.612	1.01	0.99	1.03	0.448	1.00	0.98	1.03	0.753
Sex (female = 1)	1.44	0.86	2.39	0.164	1.35	0.80	2.30	0.265	1.46	0.90	2.35	0.126	1.52	0.93	2.48	0.098
Educational background (university/graduate school = 1)	0.88	0.60	1.28	0.506	0.77	0.52	1.16	0.211	0.88	0.62	1.26	0.488	1.04	0.71	1.53	0.827
Number of children (one or more = 1)									**1.38**	**1.01**	**1.89**	**0.041**	1.00	0.68	1.46	0.990
Family caregivers (yes = 1)									0.90	0.41	1.97	0.788	0.84	0.38	1.87	0.670
Living alone (living alone = 1)									**0.59**	**0.41**	**0.86**	**0.005**	**0.61**	**0.39**	**0.94**	**0.024**
4. Well-known	*n* = 117								*n* = 40							
Age at time of survey	0.98	0.96	1.00	0.086	0.82	0.32	2.13	0.688	-				-			
Age at time of employment at facility	0.99	0.95	1.03	0.551	0.63	0.09	4.55	0.651	-				-			
Sex (female = 1)	0.93	0.52	1.69	0.824	0.87	0.46	1.62	0.658	-				-			
Educational background (university/graduate school = 1)	**2.29**	**1.50**	**3.49**	**<0.001**	**2.20**	**1.39**	**3.47**	**<0.001**	-				-			
5. Recommended by family, acquaintances, or teachers	*n* = 130								*n* = 118							
Age at time of survey	**1.02**	**1.00**	**1.04**	**0.023**	**2.95**	**1.20**	**7.26**	**0.019**	1.02	1.00	1.04	0.056	**1.03**	**1.00**	**1.05**	**0.028**
Age at time of employment at facility	0.99	0.95	1.03	0.545	0.79	0.14	4.43	0.790	0.99	0.96	1.02	0.571	0.98	0.95	1.01	0.139
Sex (female = 1)	1.82	0.92	3.63	0.087	1.88	0.92	3.84	0.082	0.73	0.42	1.28	0.270	0.69	0.39	1.23	0.211
Educational background (university/graduate school = 1)	0.87	0.55	1.37	0.545	1.05	0.64	1.71	0.853	0.95	0.60	1.53	0.845	1.13	0.69	1.87	0.631
Number of children (one or more = 1)									1.18	0.78	1.77	0.434	0.96	0.59	1.59	0.887
Family caregivers (yes = 1)									0.60	0.18	2.03	0.414	0.54	0.16	1.84	0.327
Living alone (living alone = 1)									0.81	0.50	1.31	0.385	0.82	0.47	1.45	0.502
6. Good salary	*n* = 93								*n* = 129							
Age at time of survey	**0.97**	**0.95**	**0.99**	**0.008**	0.98	0.95	1.00	0.080	**0.97**	**0.95**	**0.99**	**0.009**	**0.97**	**0.95**	**1.00**	**0.038**
Age at time of employment at facility	0.97	0.92	1.02	0.179	0.98	0.93	1.03	0.348	0.98	0.95	1.01	0.131	0.99	0.95	1.02	0.417
Sex (female = 1)	1.55	0.72	3.32	0.259	1.32	0.60	2.92	0.485	0.76	0.44	1.32	0.335	0.67	0.38	1.17	0.160
Educational background (university/graduate school = 1)	**1.86**	**1.17**	**2.97**	**0.009**	1.55	0.93	2.57	0.092	1.52	1.00	2.33	0.051	1.26	0.80	2.00	0.317
Number of children (one or more = 1)									1.01	0.68	1.49	0.973	1.50	0.91	2.48	0.114
Family caregivers (yes = 1)									0.76	0.26	2.22	0.613	0.76	0.25	2.26	0.618
Living alone (living alone = 1)									1.21	0.78	1.87	0.389	1.38	0.81	2.35	0.230
7. Convenient working style	*n* = 68								*n* = 222							
Age at time of survey	1.01	0.99	1.04	0.395	1.01	0.98	1.04	0.435	0.99	0.98	1.01	0.339	**0.96**	**0.94**	**0.99**	**0.001**
Age at time of employment at facility	**1.05**	**1.01**	**1.09**	**0.024**	**1.04**	**1.00**	**1.09**	**0.038**	**1.04**	**1.02**	**1.07**	**0.0002**	**1.06**	**1.04**	**1.09**	**<0.0001**
Sex (female = 1)	0.79	0.39	1.60	0.509	1.00	0.47	2.12	0.996	1.21	0.74	2.00	0.446	1.40	0.83	2.36	0.205
Educational background (university/graduate school = 1)	1.09	0.61	1.94	0.772	1.26	0.68	2.33	0.470	1.02	0.71	1.48	0.9008	1.00	0.67	1.49	0.9929
Number of children (one or more = 1)									**1.48**	**1.06**	**2.06**	**0.021**	**1.72**	**1.13**	**2.61**	**0.011**
Family caregivers (yes = 1)									0.90	0.39	2.06	0.795	0.97	0.41	2.26	0.938
Living alone (living alone = 1)									0.83	0.57	1.22	0.338	1.03	0.65	1.63	0.884
8. I can demonstrate my abilities	*n* = 53								*n* = 70							
Age at time of survey	**1.06**	**1.03**	**1.10**	**<0.001**	**1.07**	**1.04**	**1.11**	**<0.0001**	**1.03**	**1.00**	**1.06**	**0.022**	**1.04**	**1.01**	**1.08**	**0.013**
Age at time of employment at facility	1.03	0.99	1.08	0.181	1.02	0.97	1.07	0.493	**1.04**	**1.00**	**1.07**	**0.030**	1.02	0.98	1.06	0.312
Sex (female = 1)	0.57	0.27	1.18	0.128	0.69	0.31	1.53	0.364	**0.51**	**0.27**	**0.97**	**0.039**	0.57	0.29	1.13	0.107
Educational background (university/graduate school = 1)	1.31	0.70	2.44	0.401	**2.18**	**1.11**	**4.29**	**0.024**	**2.23**	**1.33**	**3.74**	**0.002**	**3.41**	**1.92**	**6.06**	**<0.0001**
Number of children (one or more = 1)									1.39	0.82	2.34	0.222	1.26	0.66	2.44	0.484
Family caregivers (yes = 1)									0.33	0.04	2.50	0.287	0.26	0.03	2.02	0.199
Living alone (living alone = 1)									0.96	0.54	1.73	0.904	1.10	0.55	2.20	0.797
9. I can do what I want	*n* = 124								*n* = 135							
Age at time of survey	1.00	0.98	1.02	0.750	1.01	0.99	1.04	0.192	**0.97**	**0.95**	**0.99**	**0.011**	0.98	0.95	1.00	0.075
Age at time of employment at facility	1.01	0.98	1.05	0.430	1.02	0.98	1.06	0.313	1.00	0.97	1.03	0.905	1.02	0.98	1.05	0.315
Sex (female = 1)	1.11	0.61	2.04	0.732	1.24	0.65	2.34	0.515	0.81	0.47	1.41	0.463	0.80	0.45	1.40	0.429
Educational background (university/graduate school = 1)	**1.79**	**1.18**	**2.73**	**0.007**	**2.04**	**1.29**	**3.22**	**0.002**	**1.77**	**1.17**	**2.66**	**0.007**	1.51	0.96	2.37	0.073
Number of children (one or more = 1)									0.70	0.48	1.02	0.066	1.15	0.70	1.88	0.578
Family caregivers (yes = 1)									0.51	0.15	1.71	0.276	0.56	0.16	1.90	0.349
Living alone (living alone = 1)									**1.95**	**1.29**	**2.92**	**0.001**	**1.90**	**1.16**	**3.11**	**0.011**
10. Atmosphere such as human relations in the workplace seems good	*n* = 51								*n* = 97							
Age at time of survey	0.99	0.96	1.02	0.407	0.98	0.95	1.02	0.362	**0.97**	**0.94**	**0.99**	**0.007**	0.97	0.93	1.00	0.082
Age at time of employment at facility	1.03	0.99	1.08	0.180	1.04	0.99	1.09	0.160	1.00	0.97	1.03	0.908	1.03	0.98	1.08	0.199
Sex (female = 1)	0.86	0.38	1.98	0.731	0.98	0.41	2.35	0.969	1.08	0.54	2.18	0.829	0.85	0.36	1.99	0.705
Educational background (university/graduate school = 1)	1.12	0.58	2.15	0.739	1.04	0.51	2.10	0.919	1.45	0.88	2.38	0.1411	1.05	0.52	2.13	0.8825
Number of children (one or more = 1)									0.71	0.45	1.11	0.131	2.03	0.93	4.43	0.074
Family caregivers (yes = 1)									0.26	0.04	1.95	0.191	0.48	0.06	3.63	0.475
Living alone (living alone = 1)									1.20	0.72	2.00	0.482	1.06	0.46	2.48	0.888

Note 1: bold indicates variables with *p*-values < 0.05.

**Table 5 nursrep-15-00058-t005:** Relationship between reasons for choosing workplace and personal attributes (3rd and 4th facilities).

Number of Employment Facilities	3rd Facility								4th Facility							
Number of job changes	2 (*n* = 420)								3 (*n* = 210)							
	Univariate				Multivariate				Univariate				Multivariate			
	Odds ratio	Lower 95%	Upper 95%	*p*-value	Odds ratio	Lower 95%	Upper 95%	*p*-value	Odds ratio	Lower 95%	Upper 95%	*p*-value	Odds ratio	Lower 95%	Upper 95%	*p*-value
1. Good education and career advancement system	*n* = 38								*n* = 14							
2. Good employee benefits	*n* = 55								*n* = 30							
3. Convenient location for commuting	*n* = 186								*n* = 98							
Age at time of survey	1.00	0.98	1.02	0.955	1.00	0.97	1.03	0.880	1.01	0.98	1.05	0.432	1.01	0.97	1.06	0.532
Age at time of employment at facility	0.98	0.96	1.01	0.212	0.98	0.95	1.01	0.242	1.00	0.96	1.04	0.994	0.98	0.93	1.02	0.317
Sex (female = 1)	**2.70**	**1.33**	**5.50**	**0.006**	**2.65**	**1.29**	**5.43**	**0.008**	**2.88**	**1.01**	**8.23**	**0.049**	**3.03**	**1.02**	**8.97**	**0.045**
Educational background (university/graduate school = 1)	0.91	0.57	1.45	0.679	0.89	0.53	1.48	0.650	0.78	0.40	1.54	0.4793	0.88	0.42	1.86	0.739
Number of children (one or more = 1)	1.36	0.91	2.03	0.134	1.24	0.75	2.05	0.393	**1.97**	**1.10**	**3.52**	**0.022**	1.75	0.85	3.62	0.131
Family caregivers (yes = 1)	1.72	0.71	4.18	0.229	1.77	0.71	4.39	0.219	4.23	0.86	20.87	0.077	3.66	0.72	18.64	0.119
Living alone (living alone = 1)	0.71	0.45	1.14	0.155	0.82	0.47	1.44	0.495	0.52	0.26	1.03	0.063	0.74	0.32	1.72	0.489
4. Well-known	*n* = 23								*n* = 12							
5. Recommended by family, acquaintances, or teachers	*n* = 65								*n* = 22							
6. Good salary	*n* = 65								*n* = 48							
7. Convenient working style	*n* = 149								*n* = 93							
Age at time of survey	0.99	0.96	1.01	0.244	0.98	0.96	1.01	0.309	1.02	0.99	1.05	0.213	1.02	0.98	1.07	0.301
Age at time of employment at facility	0.99	0.96	1.02	0.631	1.01	0.98	1.04	0.590	1.01	0.98	1.05	0.432	0.99	0.95	1.04	0.828
Sex (female = 1)	**8.93**	**2.72**	**29.33**	**0.000**	**9.00**	**2.72**	**29.73**	**0.0003**	2.59	0.90	7.41	0.076	2.84	0.97	8.31	0.057
Educational background (university/graduate school = 1)	1.09	0.67	1.76	0.737	0.98	0.57	1.66	0.9341	0.88	0.45	1.74	0.7197	1.08	0.52	2.26	0.830
Number of children (one or more = 1)	1.11	0.74	1.68	0.613	0.91	0.54	1.54	0.733	1.43	0.80	2.54	0.223	1.29	0.63	2.63	0.492
Family caregivers (yes = 1)	0.72	0.27	1.89	0.499	0.68	0.25	1.85	0.453	1.61	0.42	6.15	0.490	1.26	0.32	4.97	0.746
Living alone (living alone = 1)	0.62	0.37	1.02	0.062	**0.54**	**0.30**	**0.99**	**0.047**	0.75	0.38	1.47	0.397	0.92	0.40	2.09	0.838
8. I can demonstrate my abilities	*n* = 48								*n* = 36							
9. I can do what I want	*n* = 72								*n* = 44							
Age at time of survey	0.98	0.95	1.01	0.115	0.99	0.96	1.03	0.751	-				-			
Age at time of employment at facility	1.00	0.96	1.03	0.810	1.01	0.97	1.06	0.600	-				-			
Sex (female = 1)	0.60	0.29	1.25	0.173	0.56	0.26	1.19	0.132	-				-			
Educational background (university/graduate school = 1)	**2.42**	**1.39**	**4.19**	**0.002**	**2.35**	**1.28**	**4.34**	**0.006**	-				-			
Number of children (one or more = 1)	0.55	0.33	0.91	0.021	0.81	0.42	1.55	0.525	-				-			
Family caregivers (yes = 1)	0.80	0.23	2.78	0.722	0.94	0.26	3.35	0.920	-				-			
Living alone (living alone = 1)	**2.11**	**1.22**	**3.67**	**0.008**	1.83	0.94	3.55	0.074	-				-			
10. Atmosphere such as human relations in the workplace seems good	*n* = 41								*n* = 29							

Note 1: bold indicates variables with *p*-values < 0.05.

## Data Availability

Data supporting the findings of this study are available from the corresponding author upon request.

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
