# Peer review of "Changes in Workplace Choice Reasons and Individual Influencing Factors for Nurses Across Job Changes: Cross-Sectional Observational Study"

_nursrep, 2025, doi:10.3390/nursrep15020058_

Round 1

Reviewer 1 Report

Comments and Suggestions for Authors

In the abstract, the study's objective should be mentioned.

The title mentions “Changes in Workplace Choice Reasons”, but in the introduction, a different concept is used: “it is necessary to elucidate the factors associated with nurses’ turnover”. It is suggested to standardise these concepts for consistency.

Under “Setting and Sample”, it states that the study took place: “In January 2023, a web survey was conducted among 2,000 qualified assistant nurses”. What is the exact period during which the survey was conducted? Was it from the 1st to the 31st of January?

In the Methods section, under the subsection “Demographic Variables”, a descriptive analysis of the sample should be included (e.g., percentage of women and men, etc.).

The Methods chapter should also include a section on “Ethical Considerations”. This should specify the institution that issued the ethical approval and the reference number of the ethics committee approval.

In the conclusion, it would be important to further emphasise the implications of this study for nursing practice, raising awareness among hospital managers and policymakers responsible for healthcare policies. For example, the text mentions: “however, workplace selection reasons may be influenced by background factors such as gender inequality, demographic trends, and medical policies”.

It would also be beneficial to include suggestions for future research.

Author Response

Response to Reviewer 1 Comments

Thank you for taking the time to review this manuscript. We believe that our manuscript has been improved greatly on the implementation of your constructive advice and suggested modifications.

Reviewer: 1

Comments 1: In the abstract, the study's objective should be mentioned.

Response 1: Thank you for pointing this out. Accordingly, we have reviewed the wording and revised the text so that the objective of the study is clear.

Comments 2: The title mentions “Changes in Workplace Choice Reasons”, but in the introduction, a different concept is used: “it is necessary to elucidate the factors associated with nurses’ turnover”. It is suggested to standardise these concepts for consistency.

Response 2: Thank you for your helpful suggestions. The concepts of leaving and changing jobs were unclear; thus, we have standardized the explanation of the purpose of this study to use expressions related to job changes. In particular, the second paragraph of the introduction explained a review on leaving jobs without defining the concepts of leaving and changing jobs; accordingly, we have added an explanation at the beginning of the paragraph to define each concept.

Comments 3: Under “Setting and Sample”, it states that the study took place: “In January 2023, a web survey was conducted among 2,000 qualified assistant nurses”. What is the exact period during which the survey was conducted? Was it from the 1st to the 31st of January?

Response 3: Thank you for pointing this out. The survey was conducted during January 23–24, 2023. We have added the relevant information to the "Setting and Sample" subsection of the “Materials and Methods” section.

Comments 4: In the Methods section, under the subsection “Demographic Variables”, a descriptive analysis of the sample should be included (e.g., percentage of women and men, etc.).

Response 4: Thank you for your suggestion. The results of the descriptive analysis of the participants are presented in the "Descriptive Statistics of Participant’s Personal Attributes" subsection of the "Results" section. Considering that the descriptive statistics of the 721 people analyzed may be difficult to parse from Table 1, we have added a corresponding explanation to the text.

Comments 5: The Methods chapter should also include a section on “Ethical Considerations”. This should specify the institution that issued the ethical approval and the reference number of the ethics committee approval.

Response 5: Thank you for your suggestion. We had previously included this information in the Back Matter to match the journal format; we have now also included this information in the “Materials and Methods” section.

Comments 6: In the conclusion, it would be important to further emphasise the implications of this study for nursing practice, raising awareness among hospital managers and policymakers responsible for healthcare policies. For example, the text mentions: “however, workplace selection reasons may be influenced by background factors such as gender inequality, demographic trends, and medical policies”.

Response 6: Thank you for your helpful suggestions. Reviewer 2 also pointed out that a more specific explanation of the impact on practice, medical institutions, and policy was needed, and suggested including a dedicated section on policy impact. Therefore, we have added a subsection ("4.4. Implications for Individual Nurse and Healthcare Organization Practice, Policy Implications, and Future Research") to the “Discussion” section and have added a more specific explanation of the situation in several countries. Furthermore, as you have pointed out, we have revised the conclusion to emphasize the impact on individual nurses' practice and to make it more useful for hospital administrators and policy makers in charge of medical policy.

Comments 7: It would also be beneficial to include suggestions for future research.

Response 7: Thank you for pointing that out. Previously, we had listed suggestions for future research in the "Limitations and Implications for Future Research" subsection of the “Discussion” section, but we have now moved this content to "4.4. Implications for Individual Nurse and Healthcare Organization Practice, Policy Implications, and Future Research" subsection and revised it accordingly.

Reviewer 2 Report

Comments and Suggestions for Authors

Dear Authors,

I would like to thank the Editorial Board of Nursing Reports for inviting me to review the manuscript entitled "Changes in Workplace Choice Reasons and Individual Influencing Factors for Nurses Across Job Changes: A Cross-Sectional Observational Study," submitted with ID nursrep-3390928. 

Overall, this original research aligns well with the journal’s scope, particularly its focus on public health and healthcare workforce dynamics. The study provides a timely exploration of job mobility trends and influencing factors among nurses in Japan, addressing critical issues of nurse retention and career management.

Strengths:

1.     The study offers a robust analysis of workplace choice dynamics, leveraging a large, representative sample of 721 nurses with diverse professional experiences.

2.     The methodological rigor is commendable, with the use of a generalized linear mixed model and logistic regression analysis to identify associations between personal attributes and workplace preferences.

3.     The topic is highly relevant to current healthcare challenges, including workforce shortages and the need for strategic retention initiatives.

Weaknesses:

1.     The paper lacks a comprehensive discussion on the potential implications of its findings for international audiences, limiting its applicability beyond Japan.

2.     While the study identifies significant factors influencing workplace choice, it does not delve deeply into actionable recommendations for policy or practice.

3.     The manuscript would benefit from a clearer explanation of how the findings could guide interventions in regions facing severe nursing shortages.

Specific comments:

1.     Introduction: Try to expand on the global context of nursing shortages to enhance the relevance of the study for an international readership with relevant references.

2.     Methods: Please clarify the rationale behind selecting the specific personal attributes analyzed and discuss any potential biases introduced by the online survey format.

3.     Results: Provide additional interpretation of why nurses increasingly prefer non-hospital settings over time, considering broader healthcare trends.

4.     Discussion: It would be beneficial to include a dedicated section on policy implications, focusing on how healthcare institutions and policymakers can use these insights to address workforce challenges.

Recommendation:

The manuscript is well-structured and contributes valuable insights into an understudied area. However, addressing the weaknesses mentioned and incorporating the suggested revisions would improve its impact. I recommend accepting the paper after minor revisions.

Best regards, 

Reviewer

Author Response

Response to Reviewer 2 Comments

Reviewer: 2

Comments 1: Dear Authors,

I would like to thank the Editorial Board of Nursing Reports for inviting me to review the manuscript entitled "Changes in Workplace Choice Reasons and Individual Influencing Factors for Nurses Across Job Changes: A Cross-Sectional Observational Study," submitted with ID nursrep-3390928.

Overall, this original research aligns well with the journal’s scope, particularly its focus on public health and healthcare workforce dynamics. The study provides a timely exploration of job mobility trends and influencing factors among nurses in Japan, addressing critical issues of nurse retention and career management.

Strengths:

Comments 2:     The study offers a robust analysis of workplace choice dynamics, leveraging a large, representative sample of 721 nurses with diverse professional experiences.

Comments 3:     The methodological rigor is commendable, with the use of a generalized linear mixed model and logistic regression analysis to identify associations between personal attributes and workplace preferences.

Comments 4:     The topic is highly relevant to current healthcare challenges, including workforce shortages and the need for strategic retention initiatives.

Responses 1–4: Thank you for taking the time to review this manuscript. We are very pleased that you recognize the strengths of this research. We believe that our manuscript has been improved greatly on the implementation of your constructive advice and suggested modifications. For our responses to your comments, we have numbered your comments consecutively.

Weaknesses:

Comments 5:      The paper lacks a comprehensive discussion on the potential implications of its findings for international audiences, limiting its applicability beyond Japan.

Comments 6:     While the study identifies significant factors influencing workplace choice, it does not delve deeply into actionable recommendations for policy or practice.

Comments 7:     The manuscript would benefit from a clearer explanation of how the findings could guide interventions in regions facing severe nursing shortages.

Responses 5–7: Thank you for your valuable comments. Because the study was focused mainly on Japan, we have revised the manuscript to consider the impact on international readers. Specifically, we have discussed specific regional and national nurse shortages in the first paragraph of the “Introduction” section. In addition, we have revised the discussion to incorporate a more global perspective and have created a dedicated subsection ("4.4. Implications for Individual Nurse and Healthcare Organization Practice, Policy Implications, and Future Research") describing how the study results can be useful, citing specific regions and countries.

Specific comments:

Comments 8:     Introduction: Try to expand on the global context of nursing shortages to enhance the relevance of the study for an international readership with relevant references.

Response 8: Thank you for your suggestion. Accordingly, we discussed specific region and country nursing shortages in the first paragraph of the “Introduction” section.

Comments 9:     Methods: Please clarify the rationale behind selecting the specific personal attributes analyzed and discuss any potential biases introduced by the online survey format.

Response 9: Thank you for pointing this out. In the "Statistical Analyses" subsection of the “Materials and Methods” section, we have mentioned that, because there is little previous research on nurse job changes, we selected the personal attributes to use in the analysis from two systematic reviews related to job turnover. We have added an explanation of the possible bias caused by the online survey format in the "Limitations" subsection of the “Discussion” section.

Comments 10:    Results: Provide additional interpretation of why nurses increasingly prefer non-hospital settings over time, considering broader healthcare trends.

Response 10: Thank you for your helpful suggestions. We have added an interpretation of why nurses prefer non-hospital settings over time, considering broader healthcare trends, to the second paragraph of "4.1. Participant Overview" subsection in the “Discussion” section.

Comments 11:    Discussion: It would be beneficial to include a dedicated section on policy implications, focusing on how healthcare institutions and policymakers can use these insights to address workforce challenges.

Response 11: Thank you for your valuable suggestions. Accordingly, we have created a dedicated subsection ("4.4. Implications for Individual Nurse and Healthcare Organization Practice, Policy Implications, and Future Research") in the “Discussion” section, where we have provided an explanation of how the results of this study can be useful for specific regions and countries.

Comments 12: Recommendation:

The manuscript is well-structured and contributes valuable insights into an understudied area. However, addressing the weaknesses mentioned and incorporating the suggested revisions would improve its impact. I recommend accepting the paper after minor revisions.

Response 12: Thank you for your valuable comments and suggestions. We believe that the quality of the paper has improved owing to your input and that the paper will have a greater impact on the global readers. We hope that the paper is now acceptable for publication.

Round 2

Reviewer 1 Report

Comments and Suggestions for Authors

The changes made are in accordance with the request.

But the tables 1, 4, and 5 are too large. I recommend summarising the information. In Table 5, there is no explanatory text about the table before its presentation.

Author Response

Response to Reviewer’s Comments

We wish to express our appreciation for the assessment of our manuscript and insightful comments and suggestions. We believe that our manuscript has been improved greatly on the implementation of your constructive advice and suggested modifications. For clarity, all modifications to the manuscript are highlighted in red.

Reviewer:

Comments 1: The changes made are in accordance with the request.

But the tables 1, 4, and 5 are too large. I recommend summarising the information. In Table 5, there is no explanatory text about the table before its presentation.

Response 1: Thank you for the insightful review of the revised manuscript and for your comment regarding the tables. Accordingly, we have revised Tables 1, 4, and 5. For Table 1, we have added a summary in the text and changed the layout of the table to make it more compact. For Tables 4 and 5, we have summarized the information in the text. For Table 4, we have deleted the blank items. For Table 5, we moved the title to a more visible position and deleted the items that were not analyzed owning to small sample sizes to make the table more compact.

We have made every effort to address the issues raised and we feel that our manuscript has been further improved. We hope that our revisions now meet your expectations.